# Mutualism provides a basis for biodiversity in eco-evolutionary community assembly

**Gui Araujo** **, Miguel Lurgi** *

Department of Biosciences, Swansea University, Swansea, United Kingdom

* miguel.lurgi@swansea.ac.uk

## Abstract

Unveiling the ecological and evolutionary mechanisms underpinning the assembly of stable and complex ecosystems is a main focus of community ecology. Ecological theory predicts the necessity of structural constraints on the network of species interactions to allow for growth and persistence of multi-species communities. However, the mechanisms behind their emergence are not well understood. An understanding of how the coexistence of diverse species interaction types could influence the development of complexity and how a persistent composition of interactions could arise in nature is needed. Using an eco-evolutionary model, we investigate the assembly of complex species interaction networks with multiple interaction types and its consequences for ecosystem stability. Our results show that highly mutualistic communities promote complex and stable network configurations, thus resulting in a positive complexity-stability relationship. We show that evolution by speciation enhances the emergence of such conditions compared to a purely ecological assembly scenario of repeated invasions by migrating species. Furthermore, communities evolved in isolation promote a disproportionately higher complexity and a larger diversity of outcomes. Our results produce valuable theoretical insight into the mechanisms behind the emergence of ecological complexity and into the roles of mutualism and speciation in community formation.

**Data availability statement:** Computer code developed to implement the model and run simulations to reproduce the results shown in

## Author summary

Ecological communities are considerably more complex than simple collections of species sharing the same environment. The large number of ecological interactions among species drives changes in populations through time that dictate the persistence of the entire community. Most research into the mechanisms of biodiversity considers different interaction types (mutualism, competition, consumer-resource) in isolation in either ecological or evolutionary contexts. In this study, we developed a community growth model that incorporates mutualism, competition, and consumer-resource interactions and considers both ecological and evolutionary mechanisms of assembly together. We found that communities formed via evolutionary speciation can reach

this paper is available on Zenodo at:
https://doi.org/10.5281/zenodo.15058455, and
on GitHub at: https://github.com/
computational-ecology-lab/mutualism-eco-evo-
assembly.

**Funding:** This work was supported by the
Leverhulme Trust (Research Project Grant #
RPG-2022-114 to ML). The funders had no role
in study design, data collection and analysis,
decision to publish, or preparation of the
manuscript.

**Competing interests:** The authors have
declared that no competing interests exist.

higher species richness and exhibit greater proportions of mutualistic interactions than purely ecological models, resulting in more complex community structures. High levels of mutualism lead to communities more resilient to disturbances, such as the arrival of new species or sudden changes in abundances. Our research extends previous efforts by aiming to understand how evolutionary processes shape the diversity of ecological interactions and the role of these interactions in species persistence. Such knowledge is essential for preserving and restoring ecosystems in the face of growing environmental degradation.

## Introduction

Developing an understanding of the mechanisms driving the emergence and maintenance of diversity and stability in natural communities is a central aim of ecology [1–3]. Classical, as well as more recent theoretical results, have revealed that the structure of species interaction networks, which in turn drive community dynamics, have a strong influence on community persistence and assembly [4–8]. The processes influencing the formation of species interaction networks and their structure should thus provide important clues to the emergence of complex and stable communities.

Complex communities of interacting species are thought to be shaped by both evolutionary and ecological mechanisms acting together [9,10]. This raises the question of how different complex structures of species interaction networks emerge through eco-evolutionary processes (i.e. driven by both ecological and evolutionary mechanisms, even if they happen at different timescales) while maintaining species persistence and ecosystem stability. Importantly, ecological communities are made up of a suite of positive and negative interactions, such as competition, mutualism, and predation, the dynamic interplay of which can influence community persistence and stability. For example, analyses of community models have shown that high proportions of mutualistic interactions embedded into food webs promote biodiversity and stability [2,11]. However, these results are mixed in the literature, with other studies showing that strong positive feedbacks from mutualistic interactions may destabilise communities comprised of multiple interaction types, thus making a higher proportion of competition or antagonism favourable to stability [12,13]. These contrasting results partly stem from the use of different functional responses defining interactions across modelling studies. A linear Type I response tends to destabilise mutualistic dynamics by generating positive feedback loops that can lead to unbounded population growth. In contrast, nonlinear saturating responses (e.g., Type II) can mitigate these effects by introducing consumption limits, thereby dampening positive feedback and promoting stability. The influence of a multiplicity of interaction types and their proportions on community persistence and stability has also been observed in empirical systems, where empirical networks comprising a suite of structured ecological interactions have been shown to improve species persistence and biodiversity of the community when compared to random interaction compositions or interaction types in isolation [14–17].

Moreover, this multiplicity of types of ecological interactions and the deterministic and stochastic processes establishing them has been shown to influence the outcome of community assembly [3]. Recent results using model communities assembled via invasions suggest that highly mutualistic invasive species (e.g. with many mutualistic interactions) can enhance stability and biodiversity of the recipient communities when positive effects of mutualisms are controlled by saturating functional responses (e.g. Type II) [18]. In addition, priority effects, in which the order of species arrival influences the outcome of assembly, constitute a

major driver of community assembly. Random differences in colonisation history are known to cause large variations in community composition [19], which promote critical selection pressures for eco-evolutionary mechanisms to act on [20].

A next step is to understand how non-random stable interaction compositions can emerge from ecological and evolutionary processes in assembled communities. Further, given the mixed evidence on the role of mutualisms on community stability [12,21,22], a gap still remains in obtaining a better understanding of the conditions under which positive interactions can potentially disrupt community stability.

Models incorporating evolutionary processes such as speciation have been shown to generate biodiversity patterns resembling those observed in natural communities, underscoring the role of evolutionary models in the analysis of community assembly [7,10,23]. However, evolutionary assembly research has traditionally focused on food webs [24–26] and mutualistic [23,27–29] networks, without a particular focus on mixed interaction types [7,10], leaving a gap in understanding the role of speciation in promoting specific interaction compositions that contribute to biodiversity and stability.

During their process of assembly, communities grow by the addition of new species, either ecological (e.g. via immigration/invasion) or evolutionary (e.g. via speciation) [30]. New species must survive, grow, and establish themselves within the ecological context in order to become part of the community. Changes prompted by these additions can disrupt the structure of the recipient community, potentially resulting in the extinction of resident species, in a process mediated by ecological interactions. Nonetheless, it is through these assembly processes that communities are built, providing a platform for the emergence of community-level patterns while challenging their stability.

Robert May, in his seminal work [5], showed that complexity and stability tend to have a negative relationship in randomly sampled ecological networks. This result prompted the search for network structures capable of sustaining stable complexity in ecological communities [8]. Specific network structures have been shown to modify the complexity-stability relationship [6,31], and empirical systems with a large number of species tend to display low connectance (i.e. proportion of realised interactions) between species [32]. Analyses of real food webs often challenge the existence of a negative complexity-stability relation [33,34]. Even though, as argued above, the answer to this conundrum may lie in a combination of interaction types, we still lack a clear understanding of how different interactions can potentially be assembled to drive a positive complexity-stability relationship. In particular, how certain interactions compositions could arise to promote the emergence of both complexity and stability in the process of eco-evolutionary community assembly, driving an increase in species richness not followed by a corresponding decrease in connectance (the two components of complexity).

In this work, we use community dynamics models to investigate the role of different interaction types (mutualism, competition, and consumer-resource) on the assembly of complex ecosystems via repeated invasions or evolution by speciation. In our framework, interactions determine species abundances over time, driving their success and extinction. The successive addition of species to the system by speciation happens with inheritance of interactions, while invasion events are characterised by randomly connected introduced species [30]. Further, we assume strong interactions between species, yielding communities with a higher proportion of mutualisms [17]. We assess the effects of evolution on community assembly by contrasting the eco-evolutionary model with the model of ecological invasions. Additionally, we investigate how interactions shape the stability of assembled communities via a linear stability analysis.

Our findings demonstrate that selection favouring mutualisms fosters complexity by enhancing species richness without reducing network connectance. Both speciation and invasion are able to increase complexity by selecting higher proportions of mutualistic interactions, but evolution promotes a stronger increase. Additionally, mutualism with Type II saturating functional response enhances stability against perturbations by shifting the trajectories of abundances to more stable regions. We address contrasting hypotheses regarding mutualism's role in stability, showing that mutualistic interactions can destabilise purely antagonistic communities despite increased complexity. Furthermore, we demonstrate that evolved communities display a disproportionately greater complexity and diversity of evolutionary assembly outcomes compared to communities assembled with a portion of ecological invasions.

## Results

We modelled a community assembly process considering multiple types of interactions among species. We adopted a dynamical model in which species growth is influenced by these interactions. Using this framework, we simulated assembly scenarios where modelled communities grow in richness while ecological interactions are selected if they enable species persistence. Since we are interested in comparing the outcome of evolutionary assembly with that of purely ecological (i.e. via invasions) assembly, we establish two types of assembly scenarios. Assembly by evolution is driven by speciation events where new species inherit most of their interactions from their parent species with some variation. Assembly by invasion, on the other hand, is implemented via the addition of new species with randomly drawn interactions with other species in the recipient community. Invaded species are initialised with different connectance and proportion of interaction types (see Methods for a full description of evolutionary and ecological assembly). Both assembly scenarios allow for ecological selection in the composition of interactions. During each assembly event, new species are added once the recipient community has reached a steady state. The establishment of new species is determined by whether they can grow from their initial abundance in the community (i.e. whether they start with a positive growth rate), and we only consider successful establishments as 'assembly events'. After an assembly event, the community reaches a new stable equilibrium, resulting from possible extinctions and re-organisation of interactions occurring as resident species adapt to the new ecological context. We adopted Type II functional responses that enable saturation of positive interactions (consumers and mutualists) and analyse a regime of high average interaction weights (see Methods for details). The composition of interaction types is sensitive to interaction strengths and the cost of positive interactions. We analysed communities with a high strength-to-cost ratio (e.g. high $\sigma/\delta$, see Methods), and we found these communities to promote the selection of mutualistic interactions and the emergence of complexity.

### Evolution and mutualism drive the emergence of complexity in assembled communities

Across assembly time (1000 assembly events), both evolved (blue) and invaded (green) communities showed a steady increase in species richness (number of species) and a slow decrease in connectance (fraction of realised interactions from all possible ones), which resulted in an overall increase in complexity. However, evolved communities managed to retain a higher number of species and higher complexity after a comparable number of assembly events (Fig 1A, 1B, and 1C). These changes were accompanied by differences in the composition of interaction types, with evolved communities accumulating a larger fraction of mutualistic

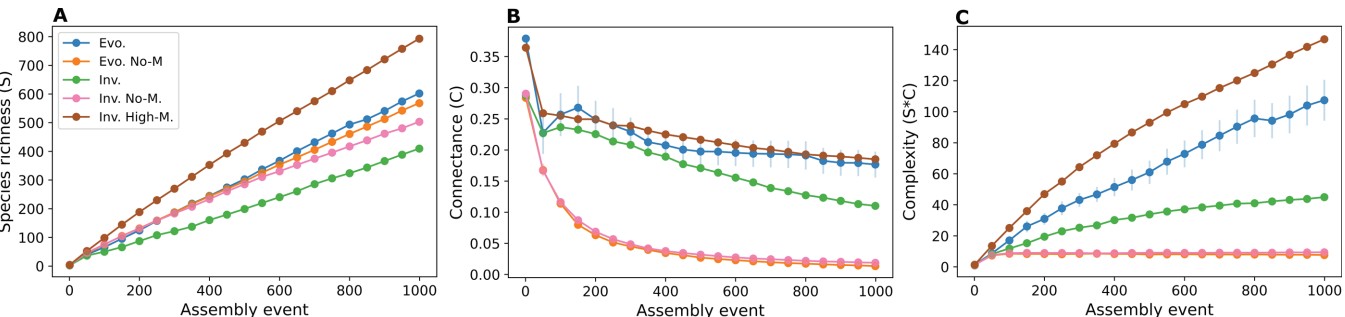

**Fig 1. Evolution shapes diversity and complexity of assembled communities.** Assembly trajectories of evolved and invaded communities. During each assembly event, a new successful (i.e. capable of growing and establishing) species is added to the community. *Evo.* scenarios are assembled via speciation, and *Inv.* scenarios are assembled via invasion. *No-M* scenarios do not feature mutualisms, while in *Inv. High-M* invading species have a proportion of mutualisms uniformly chosen between 0.8 and 1. The plots show the average values of 15 samples for every 50 assembly events (dots) and the vertical lines show the standard errors, calculated as the standard deviation divided by the square root of the number of samples. Each sample is a simulation of the entire assembly process for a scenario. Changes in richness (*S*) (A), connectance (*C*) (B), and complexity (*S*C*) (C) through time across assembly scenarios. All scenarios generate large values of species richness, however, without mutualisms (yellow and green) this comes about at the expense of connectance. Complexity (*S*C*) increases in communities with large proportions of mutualism. When these large proportions are enforced in invasion assembly (pink), complexity is high. However, when communities can select the composition of interactions (red and blue), evolution is responsible for the higher richness, connectance, and complexity observed. Parameter values: interaction strengths were drawn from a half-normal distribution of zero mean and a standard deviation of 0.2, and strength for consumers was made no larger than the strength for resources. Communities started with 5 non-interacting species. New species were given initial abundances equal to the extinction threshold $x_{ext} = 10^{-6}$. The connectance of invading species was drawn from a uniform distribution between 0.05 and 0.5. In speciation, offspring species had up to 5 interactions differing from the parent species, created or erased randomly ($\Delta = 5$, see Methods). Handling times for mutualism and consumer-resource were 0.1.

interactions, at the expense of other types such as consumer-resource and competitive ones (Fig 2A, 2B, and 2C). Given the beneficial effects of mutualistic interactions on population growth, this prevalence of mutualistic interactions also allows species in evolved communities to grow to larger abundances on average than in their invaded counterparts (Fig 2D). This suggests that evolution exploits mutualisms to enable the emergence of more complex communities comprised of more abundant species [35].

Given the strong impact of mutualistic interactions on evolutionary assembly outcomes suggested by these results, we aimed to disentangle the effects of mutualism and species formation on community assembly. We thus devised three further assembly scenarios that allow us to explore the interplay between mutualism and community growth: evolution without mutualism (orange), invasion without mutualism (pink), and invasion with enforced high mutualism (brown). In the last scenario, of invasion with enforced high mutualism, we artificially control the interaction types of introduced species to guarantee a very high proportion of mutualisms without relying on an ecological selection. This allows us to investigate whether mutualistic interactions are solely responsible for the differences we observe between evolution and invasion scenarios by looking into what happens when communities are invaded by predominantly mutualistic species.

Assembly without mutualism by either speciation or invasion produced communities with comparable species richness to that observed for evolved communities with mutualism. However, the number of interactions sustained was considerably lower, resulting in lower complexity (Fig 1A, 1B, and 1C). Evolved communities without mutualism exhibited higher proportions of consumer-resource interactions compared to competitive ones. This trend was less significant in their invaded counterparts (Fig 2B and 2C).

When high proportions of mutualistic interactions are enforced in the species introduced into invaded communities, both species richness and community complexity increase to

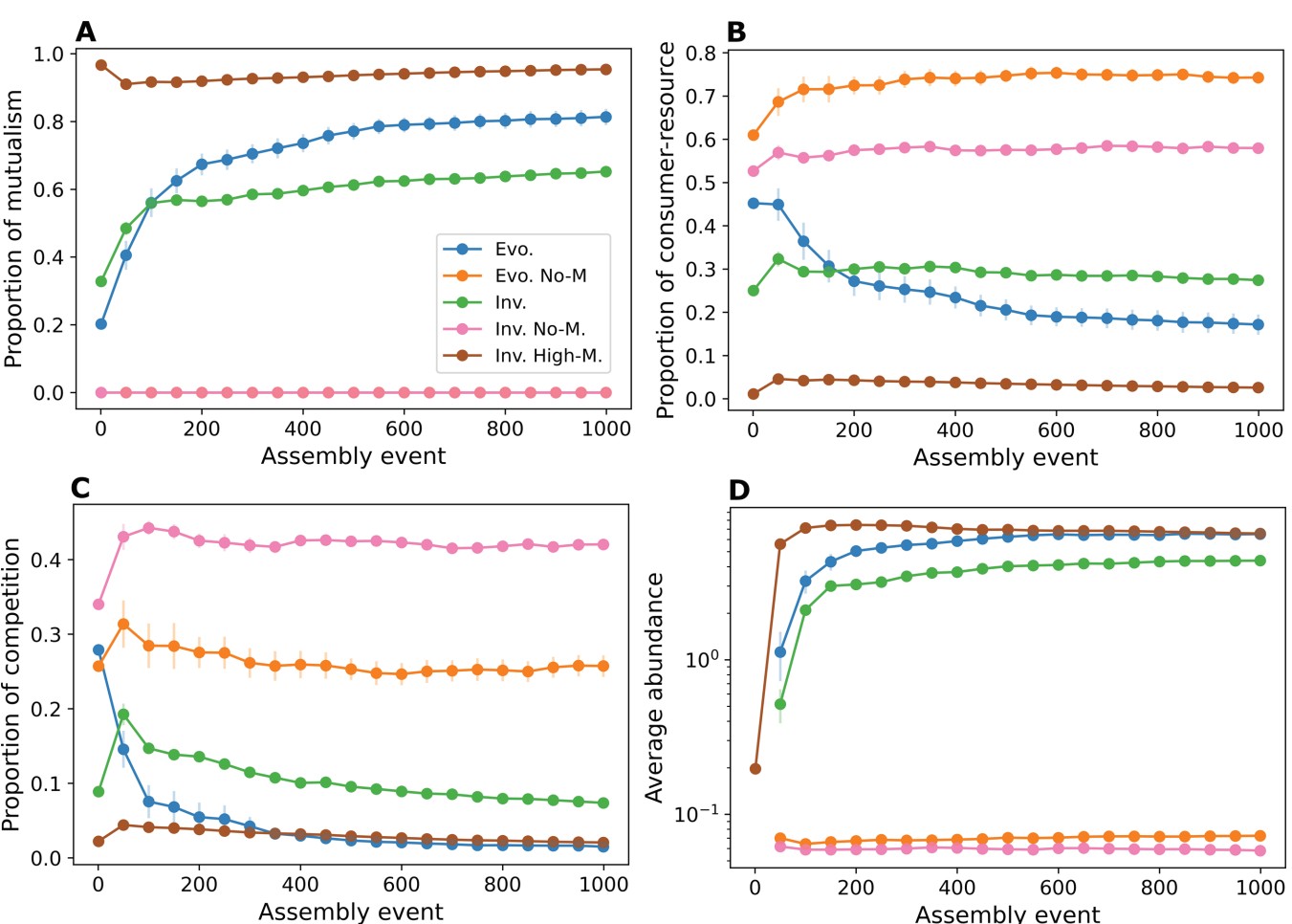

**Fig 2. Evolution promotes higher proportions of mutualism and higher abundances.** Assembly trajectories of evolved and invaded communities. During each assembly event, a new successful (i.e. capable of growing and establishing) species is added to the community. *Evo.* scenarios are assembled via speciation, and *Inv.* scenarios are assembled via invasion. *No-M* scenarios do not feature mutualisms, while in *Inv. High-M* invading species have a proportion of mutualisms uniformly chosen between 0.8 and 1. (A-C) Under both evolutionary and ecological (i.e. via invasion) assembly (red and blue), mutualistic interactions are favoured at the expense of competitive and consumer-resource ones in the composition of interaction types. In the absence of mutualisms (pink and orange curves, which then overlap at zero in A), consumer-resource interactions are dominant, but competition is not eliminated. (D) Mutualism also promotes higher average species abundances, i.e. total abundance of all species in the community divided by species richness, resulting in larger and denser communities. The plots show the average values of 15 samples for every 50 assembly events (dots) and the vertical lines show the standard errors, calculated as the standard deviation divided by the square root of the number of samples. Each sample is a simulation of the entire assembly process for a scenario. Parameter values: interaction strengths were drawn from a half-normal distribution of zero mean and a standard deviation of 0.2, and strength for consumers was made no larger than the strength for resources. Communities started with 5 non-interacting species. New species were given initial abundances equal to the extinction threshold $x_{ext} = 10^{-6}$. The connectance of invading species was drawn from a uniform distribution between 0.05 and 0.5. In speciation, offspring species had up to 5 interactions differing from the parent species, chosen randomly ($\Delta = 5$, see Methods). Handling times for mutualism and consumer-resource were 0.1.

levels comparable to evolved communities (Fig 1A, 1B, and 1C). Furthermore, the presence of mutualistic interactions also enhances average population abundances, as seen under the evolution scenario (Fig 2D). These results suggest that mutualism is a key driver of community complexity, but also that evolution acts synergistically with mutualism to enable the natural occurrence of higher levels of complexity (Fig 1C).

Despite the predominance of mutualisms in assembled communities, both evolved and invaded, they still displayed the highest average number of consumer-resource interactions per species across all scenarios explored (S1 Fig). This suggests that the presence of mutualisms fosters the presence of other types of ecological interactions, such as consumer-resource, in the community.

Both mutualistic interactions and evolutionary assembly were independently associated with a higher increase in degree entropy, measured in relation to the average value for random networks with the same connectance and richness across 50 samples (see Methods for details) (S2 Fig). A higher degree entropy translates into a more evenly distributed number of interactions per species. Communities with mutualism show a higher increase than all communities without mutualism, with the highest being for invasion with enforced high mutualism. However, evolutionary assembly, both with and without mutualism, features higher degree entropy values than the corresponding invasion assembly (S2 Fig). Lastly, we observed that the increase in network modularity (also measured in relation to random networks) was a defining feature of evolved communities, with evolutionary assembly featuring the highest values of increase both with and without mutualism (S3 Fig). Evolution thus favours the emergence of more modular communities composed of homogeneously connected species.

Our results are robust across a range of values of interaction strengths ($c_{ij}$'s, $p_{ij}$'s, and $m_{ij}$'s). Simulations with $\sigma$ values of 0.4, 0.6, and 0.8 for both *Evo.* and *Inv.* scenarios (5 samples for each combination of $\sigma$ and scenario type) yield qualitative similar results to those with $\sigma = 0.2$ (S4 Fig). Simulations with further degrees of inheritance strength (with $\Delta = \{15, 25, 35\}$ for the *Evo.* scenario, 10 samples each), are consistent with the general trend from speciation to invasion, as evolutionary assembly results approach the ones of invasion assembly for lower inheritance degrees (S5 Fig). Additionally, assuming speciation rates to be proportional to abundance in assembly by evolution, instead of uniform, did not qualitatively altered our results (S7 Fig).

## Mutualism can enhance or disrupt stability and complexity

To understand the effects of evolution and mutualism on the stability of complex communities, we conducted a linear stability analysis to evaluate the stability of assembled communities against external perturbations. This analysis revealed that the distribution of eigenvalues of the community matrix at the end of simulations (see Methods) is influenced by the proportion of mutualistic interactions. The presence of a large proportion of mutualism, both under ecological and evolutionary assembly, considerably shifts the distributions to the negative, 'more stable' zone of the eigenvalue space (Fig 3). A Type II functional response might be relevant in producing this result. Moreover, evolution produces a concentration of eigenvalues on the real axis, which is associated with less oscillatory behaviour of the system dynamics. This suggests that mutualistic interactions are strong drivers of stability in complex communities [21], and that evolution plays a role in generating more steady equilibrium states.

To further investigate the impact of mutualism on the assembly process, we designed two additional scenarios of assembly by invasion. One scenario starts with mutualism, then halfway through the simulation (i.e. after 500 assembly events) mutualistic interactions stop appearing in new invading species. In the second scenario, simulations start without the presence of mutualisms, then halfway through the simulation mutualistic interactions are allowed to appear via new invading species.

When mutualistic interactions are allowed at the beginning of the assembly process, they almost instantly dominate the system, with their proportion decreasing after mutualisms are

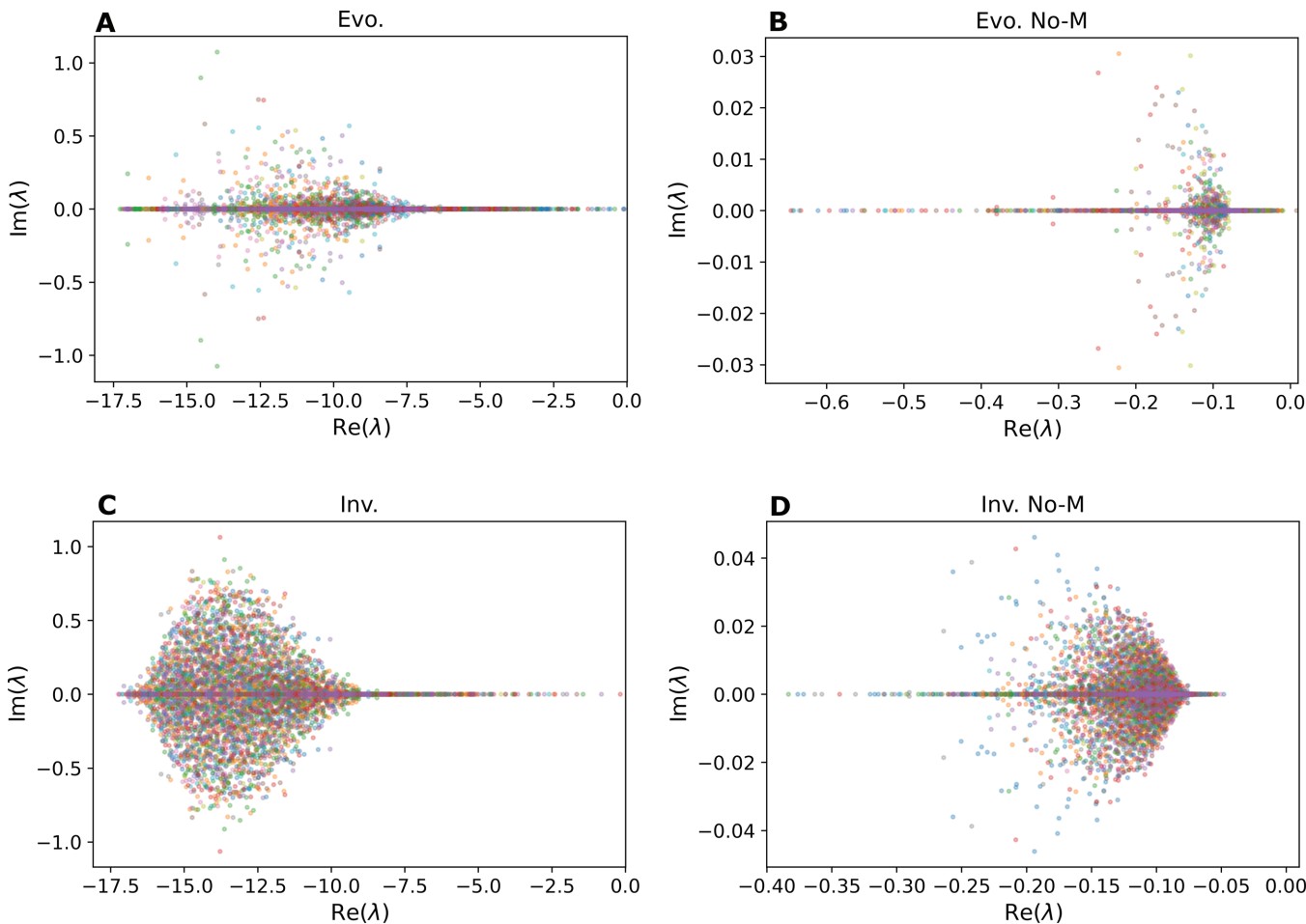

**Fig 3. Mutualistic interactions enhance the dynamical stability of assembled communities.** Distribution of real and imaginary parts of all the eigenvalues of the community matrix ($\lambda$) in the stability analysis of assembled scenarios (see Methods for details). As long as the real part of every eigenvalue is negative, the system is stable. Points with different colours represent each of the 15 different samples of assembled networks (i.e. outcomes of different simulations). *Evo.* scenarios are assembled via speciation, and *Inv.* scenarios are assembled via invasion. *No-M* scenarios do not feature mutualisms. High proportions of mutualism produce distributions of eigenvalues with a large shift to the left of the plane, which translates into communities being more resistant to external perturbations. Evolution produces a pattern of eigenvalues with smaller imaginary parts (i.e. closer to the x axis), which is related to a less oscillatory behaviour of the population dynamics. (A-D) Respectively for: evolution, evolution without mutualism, invasion, and invasion without mutualism. Parameter values: interaction strengths were drawn from a half-normal distribution of zero mean and a standard deviation of 0.2, and strength for consumers was made no larger than the strength for resources. Communities started with 5 non-interacting species. New species were given initial abundances equal to the extinction threshold $x_{ext} = 10^{-6}$. The connectance of invading species was drawn from a uniform distribution between 0.05 and 0.5. In speciation, offspring species had up to 5 interactions differing from the parent species, chosen randomly ($\Delta = 5$, see Methods). Handling times for mutualism and consumer-resource were 0.1.

turned off and new invaders are not allowed to carry mutualistic interactions, as expected (Fig 4A). This, however, does not have a significant impact on species richness, which keeps increasing as new species are added to the system, although it prompts a decrease in network connectance and complexity (Fig 4B, 4C, and 4D). Unexpectedly, the sudden incorporation of mutualistic interactions in the scenario where they are not allowed since the beginning causes a major disruption in species richness (Fig 4B). Despite this disruption, the effect on connectance and complexity is opposite to that observed in the previous scenario, both undergoing a large increase once mutualisms are brought into the assembly process (Fig 4C and 4D).

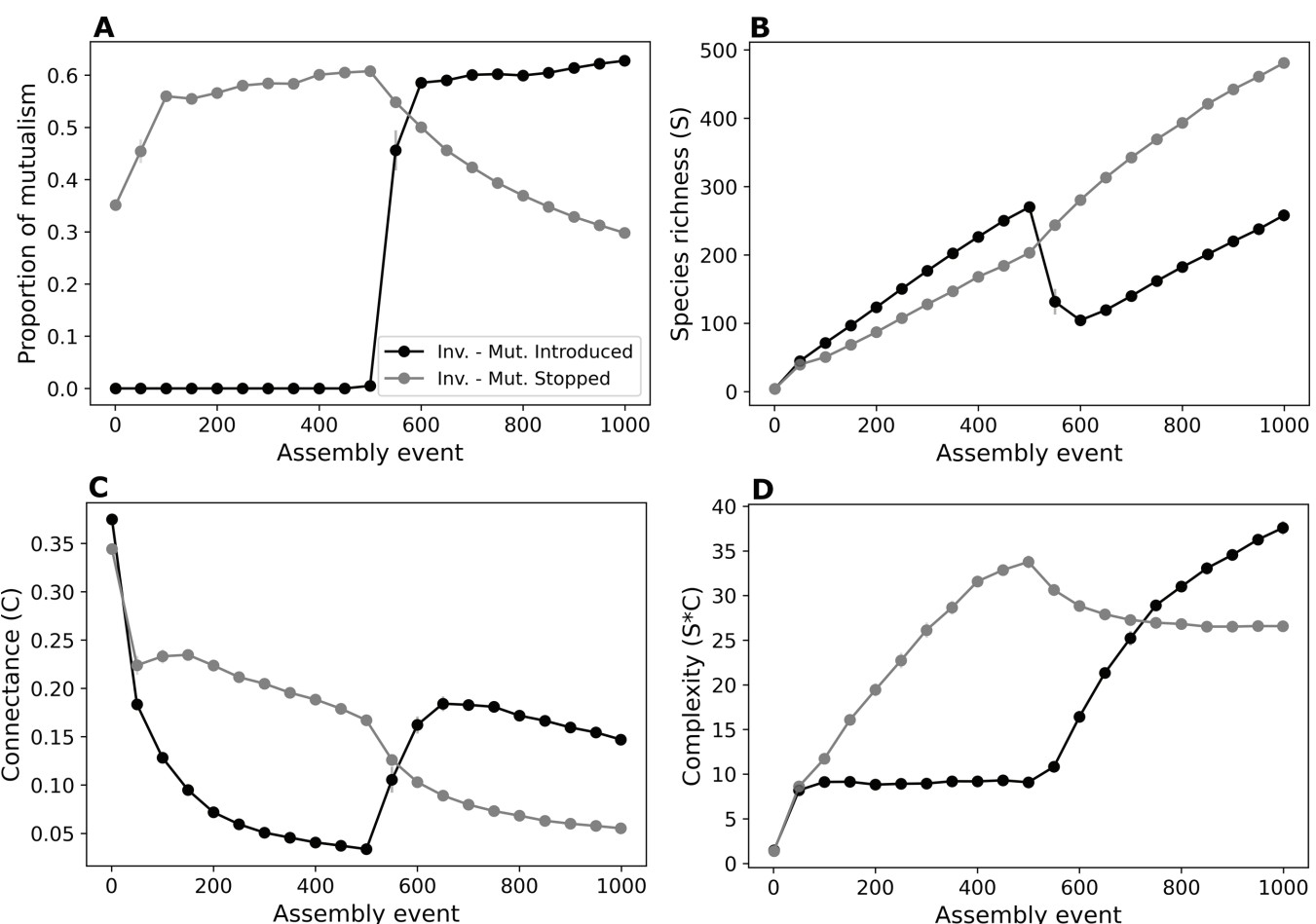

**Fig 4. The introduction of mutualisms into assembled communities increases their connectance and complexity while decreasing their richness.** Using the invasion model, we investigate the effect of switching on and off (black vs grey) invasions with mutualisms halfway through the simulation (i.e. after 500 assembly events). (A) When mutualistic interactions are allowed from the beginning of the assembly process, they quickly rise in proportion and dominate the community. When they stop being introduced in further assembly events (i.e. introduced species do not carry any mutualistic interactions), their proportion slowly decreases with successive invasions. (B) Even though higher proportions of mutualism promote higher richness, introducing this type of interaction into already assembled large communities promotes a sudden drop in richness, while stopping mutualism promotes a slight boost in richness increase. (C) Mutualism also promotes an increase in network connectance when introduced into assembled communities, while stopping mutualistic interactions from entering an assembled system slowly decreases it. (D) As a result, the introduction of mutualistic interactions promotes a growth in complexity in communities where it was once established as low, while stopping the introduction of further mutualistic interactions causes a slight decrease in complexity. The plots show the average values of 15 samples for every 50 assembly events (dots) and the vertical lines show the standard errors, calculated as the standard deviation divided by the square root of the number of samples. Each sample is a simulation of the entire assembly process for a scenario. Parameter values: interaction strengths were drawn from a half-normal distribution of zero mean and a standard deviation of 0.2, and strength for consumers was made no larger than the strength for resources. Communities started with 5 non-interacting species. New species were given initial abundances equal to the extinction threshold $x_{ext} = 10^{-6}$. The connectance of invading species was drawn from a uniform distribution between 0.05 and 0.5. In speciation, offspring species had up to 5 interactions differing from the parent species, chosen randomly ($\Delta = 5$, see Methods). Handling times for mutualism and consumer-resource were 0.1.

The incorporation of mutualisms removes an apparent upper bound on the value of community complexity (Fig 4D), suggesting that complexity is strongly controlled by the presence and proportion of mutualistic interactions.

### Evolution as the fundamental driver of community complexity

Ecological communities are likely to be assembled via a combination of ecological (i.e. invasions) and evolutionary (i.e. speciation) processes. To unveil the effects of a mixture of ecological and evolutionary processes in determining the outcome of community assembly, we investigated assembly at the interface between invasion and evolution. We devised a set of assembly scenarios ranging from pure evolution to pure invasion, including intermediate versions varying the proportion (i.e. probability) of evolutionary versus invasion assembly events across three values: 20%, 50%, and 80%.

Results from these assembly scenarios revealed that species richness decreases when shifting from pure evolution to pure invasion. The decrease in species richness was generally proportional to the increase in the probability of invasion events, however the scenario with 80% of invasion was closer to pure invasion than expected by the proportionality (Fig 5A). The same pattern was present in the proportion of mutualistic interactions, but even stronger, with no apparent increase from pure invasion to 80% invasion (Fig 5B).

Network connectance, on the other hand, showed a disproportionate increase for pure evolution in comparison to every scenario with some proportion of invasions (Fig 5C). Consequently, an even larger difference from pure evolution to the other scenarios resulted for complexity. A mixture with majority of evolution (80%) showed a higher value of complexity than expected in a proportional increase, and pure evolution showed an even wider gap in the value of complexity (Fig 5D). This highlights evolution as an underlying mechanism for the emergence of network complexity in complex communities.

Not only the pure evolutionary model seemed to promote higher connectance, complexity, and proportion of mutualistic interactions than the mixed models, but it was also capable of generating a larger variety of outcomes in assembled communities (Fig 6). This further suggests that the isolated evolution of communities, assembled in the absence of invading species, has an additional tendency for diversity. In purely invaded communities, connectance decreased as species richness increased but this trend was alleviated as the proportion of evolutionary assembly events increased (Fig 6). This result shows that an increase in speciation events relative to invasions reverses the negative relationship between species richness and connectance in assembled communities.

## Discussion

A diversity of interaction types has been shown to enhance persistence and stability in complex ecological communities, both within assembled networks [2,11] and throughout the process of community assembly [18]. However, the evolutionary assembly (i.e. via speciation) of communities with multiple interaction types, and its relation to ecological assembly via invasions remains unexplored. Specifically, we lack a description of how these two processes (ecology and evolution) might lead to distinct structures and compositions of interaction types, ultimately influencing complexity at the community level. Assuming a scenario dominated by strong interactions, we investigated how community assembly dynamics influence interaction diversity, with a particular focus on the role of evolution in promoting stability while generating complexity and biodiversity.

Our findings reveal at least three important lessons for community assembly. First, when compared with assembly via invasion, speciation generates communities with higher species richness and connectance, which leads to significantly higher complexity. The mechanism underlying this difference is a stronger selection for higher proportions of mutualistic interactions. Second, the assembly of highly mutualistic communities not only enhances complexity

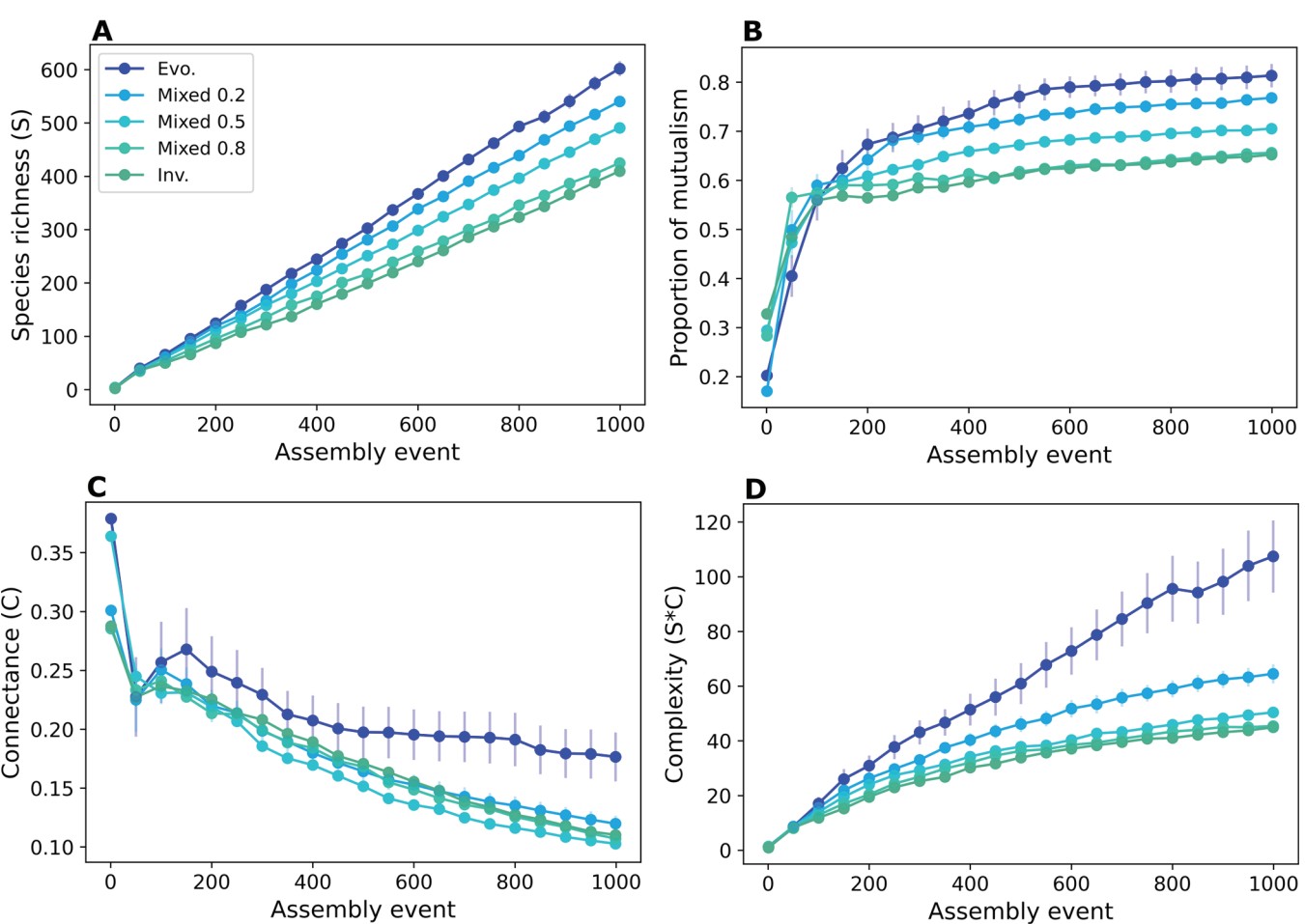

**Fig 5. Evolutionary assembly drives community complexity.** With a mixture of ecological (i.e. via invasion) and evolutionary (i.e. via speciation) assembly events, we investigate the effect of the combination of invasion and evolution on the process of community assembly. (A) The scenario of pure invasion (green) promotes the smallest values of richness while pure evolution (blue) promotes the highest values. Mixed scenarios, with 20%, 50%, and 80% of invasion, exhibit a generally proportional decrease in values of richness from pure evolution to pure invasion. Except that pure invasion and 80% invasion are closer together than expected in a proportional difference. (B) The proportions of mutualistic interactions follow a similar pattern. (C) A disproportional increase of connectance happens in the scenario of pure evolution, while all scenarios containing invasion promote values of connectance that are closer together. (D) As a result, pure evolution promotes a disproportional increase in complexity. A majority of evolution also exhibits a gap in complexity compared to the scenarios with 50% and 80% of invasions and the scenario of pure invasion. The plots show the average values of 15 samples for every 50 assembly events (dots) and the vertical lines show the standard errors, calculated as the standard deviation divided by the square root of the number of samples. Each sample is a simulation of the entire assembly process for a scenario. Parameter values: interaction strengths were drawn from a half-normal distribution of zero mean and a standard deviation of 0.2, and strength for consumers was made no larger than the strength for resources. Communities started with 5 non-interacting species. New species were given initial abundances equal to the extinction threshold $x_{ext} = 10^{-6}$. The connectance of invading species was drawn from a uniform distribution between 0.05 and 0.5. In speciation, offspring species had up to 5 interactions differing from the parent species, chosen randomly ($\Delta = 5$, see Methods). Handling times for mutualism and consumer-resource were 0.1.

but also contributes to stability. However, mutualism can also act as a destabilising force, triggering an extinction wave when introduced through invasions into antagonistic communities. Finally, pure evolutionary assembly promotes a disproportionately greater increase in complexity when compared to mixed assembly comprising speciation and invasion acting in tandem. Interestingly, speciation reverses the negative relationship between species richness and network connectance, creating greater variability in outcomes for richness and connectance distributions and, consequently, greater variability in complexity.

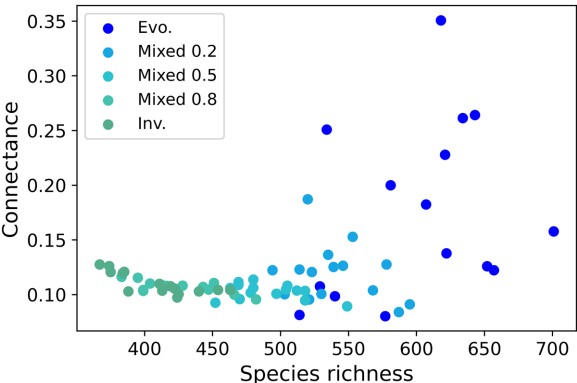

**Fig 6. Evolutionary assembly drives the diversity of assembly outcomes.** A combination of connectance and species richness in assembled communities (i.e. at the end of simulations) reveals that the presence of evolution promotes a larger variation of possible outcomes of community size and connectivity. In the plot, each point represents the final outcome of a single sample after 1000 assembly events, with mixed models labelled following their proportion of speciation events (e.g. Mixed 0.2 means 20% of invasion and 80% of speciation events). Each sample is a simulation of the entire assembly process for a scenario. Purely evolutionary assembly trajectories, as well as those with a majority of evolutionary events compared to invasions, result in a disproportionately higher diversity of outcomes. Increasing proportions of evolutionary assembly events result in the increasing capacity to grow in richness without the cost of decreasing connectance. As communities move from pure invasion to pure evolution, a negative relationship between richness and connectance is gradually reversed. Parameter values: interaction strengths were drawn from a half-normal distribution of zero mean and a standard deviation of 0.2, and strength for consumers was made no larger than the strength for resources. Communities started with 5 non-interacting species. New species were given initial abundances equal to the extinction threshold $x_{ext} = 10^{-6}$. The connectance of invading species was drawn from a uniform distribution between 0.05 and 0.5. In speciation, offspring species had up to 5 interactions differing from the parent species, chosen randomly ($\Delta = 5$, see Methods). Handling times for mutualism and consumer-resource were 0.1.

Our evolved communities revealed a wide range of complexity values, with complexity being associated with mutualisms. This relationship implies that evolved mutualism enables the generation of a range of different combinations in values of species richness and connectance. Chomicki et al (2019) [36] reviewed the variety of roles that different types of mutualism can assume in impacting species richness, which can differ radically depending on the biological details of interactions and the environmental conditions. These findings suggest that further detailing the biology of mutualistic interactions can reveal patterns underlying the variation we found in the outcomes of species richness and connectance. Further exploration of different types of mutualistic interactions could thus reveal a more precise relationship between the types of mutualism, evolution, and community complexity. This could be achieved, for example, by considering the addition of species traits into the models to distinguish the types of mutualism, evolving these traits using frameworks akin to adaptive dynamics [37]. The food web literature, with its rich focus on species traits [38–40], offers a valuable framework for extending this analysis. Beyond the incorporation of trait adaptation mechanisms to complement our treatment of evolution as speciation events, other evolutionary mechanisms such as adaptive foraging can further promote the emergence of complexity in these model ecosystems [41].

Communities formed through a strong contribution of speciation events, often by the understudied process of adaptive radiation, are found on remote oceanic islands [42]. Across different islands, these communities are characterised by generalised mutualistic interactions and the presence of super-generalist species, and these interactions are fundamental to the maintenance of biodiversity and the weakening of latitudinal diversity gradients [43,44].

Despite their small area, these islands are responsible for a disproportionately high biodiversity globally [45]. These features are recapitulated in our results for communities assembled mainly by speciation events, which present a stronger selection for widespread mutualistic interactions, and these interactions in turn promote the emergence of more complex and diverse communities. Our results thus generate testable hypotheses for the mechanisms driving observed patterns of diversity in oceanic islands, a promising area of future research.

A possible explanation for why evolution produces a larger variety of outcomes in communities than assembly by invasion is that the structure in the distribution of interactions can be developed and greatly preserved by inheritance, which might be seen as the formation of a diversity of specific lineages. Without inheritance, such a structure can not be maintained in the face of an influx of randomly generated interactions. Additionally, the inheritance of lineages might act as a mechanism by which modularity increases in evolved communities in our simulations. Previous results using an adaptive niche model showed that an increase in modularity is a consequence of adaptation in mutualistic communities [27], where networks were rewired into an adaptive optimal state with increased modularity. Our results complement these findings by showing the emergence of increased modularity in mutualistic networks as communities grow by speciation.

Another mechanism with the potential to impact community structure is the rapid evolution of ecological interactions, in which evolutionary changes happen at ecological timescales [46,47]. Therefore, the addition of new species during the ecological transient dynamics, before equilibrium, might result in different community assembly patterns. Additional simulations with several rates of out-of-equilibrium assembly events revealed similar results to those observed in our main simulations (S6 Fig). However, speciation events occurring at high frequency resulted in a significant loss of interactions and a consequent drop in the values of community complexity. These results suggest that rapid evolution might strongly impact ecological interactions and the complexity of ecological systems. Given the empirical relevance of rapid evolutionary changes, this phenomenon deserves further investigation in future studies.

In our simulations, high proportions of mutualistic interactions increase community stability while mutualism destabilises a community dominated by competition and consumer-resource interactions. These findings are in agreement with previous predictions that mutualism enhances stability only if competition levels are low, otherwise it decreases stability [48]. Previous models by Mougi and Kondoh (2012, 2014) [2,49] also agree with these results, and our analysis extends their work by including community assembly, where these results are obtained as an emergent property. In contrast to them, however, we do not assume constant interaction effort, which constrains the effects of positive interactions by further dividing them by the sum of positive interaction strengths for each species. Our results suggest that a high proportion of mutualistic interactions can promote a positive complexity-stability relationship [50] without the constant effort assumption, differing from the previous suggestion by Suweis et al. (2014) [51] that a positive relationship depends on this assumption. Our use of Type II functional responses implementing the saturation of interactions might explain why mutualisms increase stability without any explicit constraints in interaction weights. The stabilisation of mutualistic interactions through the use of Type II functional responses points to an important reason for the apparent contradiction between our results and those from previous studies, which vary in assumptions and modelling frameworks used. For example, Coyte et al. (2015) [12] concludes that mutualisms are destabilising, but their model uses a Type I functional response that generates positive feedbacks between mutualistic species. Maliet et al. (2020) [52] conclude that mutualisms reduce diversity, but they use an individual-based model with fixed population size and interactions between explicitly defined

individuals. We argue that a way forward to address apparent contradictions between mod-elling approaches is to map the different conditions and assumptions made by different mod-els into a bigger picture of distinct scenarios with potentially different results. This will pro-vide clarity regarding the conditions under which different interaction types can stabilise or destabilise system dynamics.

The destabilising effect of low proportions of mutualistic interactions becomes more nuanced when the assembly process is considered. Following a wave of extinctions, mutual-ism sharply increases its proportion and the community starts growing again. In this process, the inclusion of mutualism always increases community complexity. This happens because the transient decline in richness is accompanied by an increase in connectance. Therefore, the assembly process reveals that the destabilising effects of mutualism are followed by a reorgan-isation that enhances biodiversity in the long term. Moreover, our results show that high pro-portions of mutualism can facilitate the coexistence of diverse interactions, supporting com-munity growth through increased species abundance and increased consumer-resource inter-actions per species compared with scenarios without mutualism. We suggest that mutualism acts as a web of widespread mutual benefits among species, maintaining higher abundances in the face of negative interactions, while the control of positive feedbacks is responsible for smaller temporal variations in abundance distributions. This feedback control, realised by a Type II functional response, also contributes to biological realism by implementing satura-tion periods where individuals do not seek positive interactions (e.g. handling and satiation times). The resulting shift of abundances to a more stable dynamical regime, as shown in our stability analysis, might be a direct consequence of such drivers of abundance distributions [21].

The feedback control given by saturating interactions results in mutualisms promoting an increased internal stability of communities [18]. However, it suggests an approach to how we define community complexity that differs from the concept put forward by classical ran-dom matrix analyses [5,6]. The classical definition of complexity includes species richness and connectance, and it also includes a measure of the overall strength of interactions in the network. This measure of strength considers effective interactions at equilibrium by directly looking at the community matrix (i.e. the system's Jacobian). In Lotka-Volterra models featur-ing linear functional responses, interactions do not saturate and increase proportionally with species biomass. When interactions saturate, an increased connectance results in effective interactions being smaller than the nominal strength of each interaction (i.e. represented by the interaction coefficients). Under these conditions, if a community is large in size and highly connected, and if species pairs interact strongly, then assembled communities are expected to be complex regardless of how the saturation of interaction benefits is realised. In real com-munities, the saturation of interactions does happen [53,54], and real community complexity does not imply species more intensely interacting with every partner. It implies the potential or capacity to interact. In our model, we implement strong pairwise interactions, and there-fore we analyse the increase in complexity by looking at richness and connectance alone. In this way, the saturation of interactions at equilibrium does not matter for complexity, only the size and the potential of the community for interactions. As such, this measure of com-plexity accurately captures the size and interactivity of the community, as it should, and it also accommodates the saturation of interactions.

In community assembly models, the addition of species by invasion implies the existence of colonisation from external sources which usually results from invading species arriving via dispersal from other connected communities. Even though our results suggest that evolution alone can serve as a mechanism underlying the increase in community complexity and diver-sity, a more explicit consideration of spatial colonisation processes such as dispersal in models

like ours would reveal further insights into the interplay between local and regional processes in community assembly. Further research focusing on investigating the joint assembly of several communities connected by dispersal as a meta-community [55] can address this gap. As such, we might ask how different patterns of dispersal or degrees of isolation can affect the diversity and the emergence of complexity in evolutionary community assembly.

The specific ways in which complexity can affect community stability depend on several considerations, one of them being the structure of ecological interactions and how they can be assembled in nature. Here we have shown that evolutionary assembly considerably impacts the emergence of complexity, diversity, richness, abundance, and network modularity of model communities, supporting predictions that networks should be sensitive to evolutionary constraints [10,23]. Our work advances the understanding of the mechanisms behind community assembly by exploring how communities assembled by evolution differ from purely ecological communities assembled via invasion events and how can these processes shape interaction-type composition and facilitate the emergence of complexity as stability. We showed that mutualistic interactions are fundamental to maintaining biodiversity and that high proportions of them have stabilising effects on the community. This knowledge is essential for devising ecosystem management and conservation aiming at enabling the emergence of robust ecosystems capable of withstanding the challenges of a changing world [56].

## Methods

### Model

**Ecological dynamics.**  To study the emergence of stable complex communities, we used a model of community assembly where the abundance of each species is governed by a system of differential equations. We defined a generalised Lotka-Volterra network where species are represented as nodes and ecological interactions between them are represented as weighted directed links. The model considers a mixture of interaction types, allowing for different combinations of signs and magnitude of interactions. Time-varying abundances are thus governed by the following set of differential equations:

$$\frac{dx_i}{dt} = x_i\Big(r_i - \sum_{p^+,m}\delta - s_i x_i + \sum_{j=0}^{S}\Big(-c_{ij}x_j - \frac{p_{ij}^- x_j}{\big(1 + h_p\sum_k p_{jk}^+ x_k\big)}\Big)\Big)$$
$$+ x_i\sum_{j=0}^{S}\Big(\frac{p_{ij}^+ x_j}{\big(1 + h_p\sum_k p_{ik}^+ x_k\big)} + \frac{m_{ij}x_j}{\big(1 + h_m\sum_k m_{ik}x_k\big)}\Big), \tag{1}$$

where $x_i$ density of species $i$ among $S$ species in the community. Parameters $p_{ij}$, $m_{ij}$, and $c_{ij}$ are non-negative and represent consumer-resource (+/−), mutualistic (+/+) and competitive (−/−) interactions respectively. These coefficients determine the strength of the effect of species $j$ on the density of species $i$. We thus considered three different types of interactions. We represent $p_{ij}^+$ if $i$ is a consumer and $p_{ij}^-$ if $i$ is a resource.

The model assumes Type II functional responses to account for the saturation of intake of mutualists and consumers. For competition, there is already an inherent saturation, since it implies a mutual loss of biomass. Consumers and mutualists have their interactions saturated by handling and satiation periods. Thus, within the functional response, $h_p$ and $h_m$ are the average saturation times for consuming and mutualistic interactions. Consumers ($p_{ij}^+$) and mutualists ($m_{ij}$) are saturated by how much they can consume, while resources ($p_{ij}^-$) are saturated by the consumer's capacity to interact (hence the subtly different denominators between consumer and resource).

In addition to ecological interactions, species have an independent growth rate $r_i$ and intraspecific competition with strength $s_i$. We implemented a cost $\delta$ that penalises the growth rate for every positive interaction that benefits the species, which means that the sum $\sum_{p^+,m}$ counts the number of mutualistic interactions and the number of interactions in which species $i$ is a consumer. We assumed the same cost $\delta = 0.01$ across all interactions. This cost $\delta$ introduces an evolutionary trade-off where the resulting independent growth rate becomes smaller and eventually negative if a species accumulates numerous positive interactions, thus increasing dependency on its positive interactions.

If the abundance of a species declines below a threshold $x_{ext}$, the species is removed from the dynamics and is considered extinct. In this process, if a species becomes decoupled from the network, without any interactions with other species, it is also removed.

**Assembly dynamics.**   Whenever the system in Eq 1 reaches equilibrium, a new species is introduced. We termed the introduction of new species as an assembly event. The new species is proposed with a set of interactions chosen according to specific rules for evolution or invasion and, if it is able to grow from the extinction threshold $x_{ext}$, it is introduced with abundance equal to $x_{ext}$. If it is not able to grow, it is rejected and another species is proposed.

**Assembly by evolution.**   Evolution in our model occurs via speciation with inheritance of interactions. In an assembly event, a parent species is randomly chosen from the community. A new offspring species is thus created and inherits the parent's interactions, with some variation. Variations are determined randomly with a mutation strength of $\Delta$, defined as the maximum difference of interactions, which is caused by removing inherited interactions and creating new ones.

Whenever a new offspring is proposed, with interactions inherited from the parent species, the number of different interactions is randomly chosen from the integer interval $[1, \Delta]$. Then, a random number of interactions is removed and created in a way in which the sum of creations and removals amounts to the chosen difference. For example, if $\Delta = 5$, an offspring can differ from the parent by a number of interactions randomly chosen from $\{1,2,3,4,5\}$. If the chosen value is 3, then one random possibility is to have 1 interaction removed and 2 interactions created, resulting in a difference of 3. This rule is versatile in terms of how much difference mutations can produce, and it also lets us fix how controlled the evolutionary process is. A high $\Delta$ means a loose evolutionary process, with less precision of inheritance. Once the difference is obtained, which interactions are removed or created is chosen randomly. In our simulations, to study the effects of strong inheritance, we kept a constant value of $\Delta = 5$.

**Assembly by invasion.**   A new species included by invasion, instead of evolution, differs in how its interactions are chosen. Invading species do not inherit interactions from a parent species, and instead have interactions randomly chosen. Interactions are assigned with a probability that is randomly set between $[\rho_1, \rho_2]$, which means a randomly defined connectance for the invading species. This allows for the connectance of the community to adapt as species with different numbers of connections can invade and be ecologically selected. We chose $\rho_1 = 0.05$ and $\rho_2 = 0.5$, reflecting a range that is compatible with the results emerging from the evolutionary assembly. The proportions of interaction types are randomly chosen for each new species, with uniform probability, allowing for the adaptation of proportions as selected by the ecological dynamics. Thus, the model is sensitive to selection for interaction types, since invading species can arrive with any configuration of interaction types. After the proportions are chosen, the strengths are chosen in the same way as with evolution, but the signs are fixed

by the determined proportions of interaction types. For the high mutualism model (*Inv. High-M*), the proportions of interactions are biased towards mutualism, with a value between 0.8 and 1 being chosen with uniform probability for mutualism proportion, and the rest being equally distributed to competition and consumer-resource. The model with no mutualism (*Inv. No-M*) sets the proportion of mutualism to 0.

**Mixed models.**   To investigate the transition between evolution and invasion and the effect of their mixture, we constructed a set of mixed models (Fig 5). In these models, we chose either evolution or invasion when an assembly event happens and new species will be proposed and introduced in the community. The label *"Mixed 0.2"* refers to a proportion of 0.2 of assembly events being of invasion and 0.8 of being of evolution, and the same for the other models, with invasion proportions of 0.5 and 0.8.

## Simulations

The simulations start with 5 non-interacting species with initial abundances drawn from the uniform distribution on the interval [0,0.02]. We used the extinction threshold $x_{ext} = 10^{-6}$ for all simulations. Simulations end after a total of 1000 assembly events. Assembly happens at equilibrium, which we consider to be when the relative changes in all abundances are less than 0.01% between successive timesteps. The intrinsic growth rates $r_i$ are chosen from a normal distribution $\mathcal{N}(\mu_r, [0.1\mu_r]^2)$, with $\mu_r = 0.1$. The intraspecific competition rates are chosen as the inverse of a lognormal distribution $s_i \propto [log\mathcal{N}(0.1, 0.5^2)]^{-1}$, where 0.1 is the mean and 0.5 is the standard deviation. Both rates are not inherited, to reflect the fact that they vary depending on the particular condition of each species in relation to the environment. The interaction saturation times were kept constant as $h_m = h_p = 0.1$. The strength of newly created interactions is randomly drawn from a normal distribution $|\mathcal{N}(0, \sigma^2)|$, with the standard deviation $\sigma$ determining how strong the interactions can be. Evolution allows for a variation in interaction strengths, with the inherited strength changed by a small noise $w' = w + \mathcal{N}(0, [0.05w]^2)$, where $w'$ is the offspring's strength and $w$ the parent's strength. For consumer-resource interactions, whenever the benefit to the consumer becomes larger than the amount of resource consumed, we made it equal. Therefore, if randomly $p_{ij}^+ > p_{ji}^-$, we set $p_{ij}^+ = p_{ji}^-$. This condition is introduced to prevent the biomass conversion efficiency for consumers from exceeding 1. After 20 assembly events, disconnected species were removed after each assembly event.

We used the function odeint from Python library Scipy [57] for numerical integration of Eq 1.

## Analysis

To analyse the histories of assembled communities, we ran 15 samples for each scenario and evaluated variables at every 50 assembly events, calculating their mean and standard error (standard deviation divided by the square root of the number of samples), represented in the graphics. The standard error indicates the uncertainty in our estimations of averages from our sets of samples.

**Network metrics.   Degree entropy:** The entropy of the network's degree distribution $P(k)$, where $k$ is the degree of nodes (number of interactions of a species). We calculated the degree entropy of the unweighted network, considering only the presence of links, as $H = -\sum_k P(k)ln(P(k))$.

**Modularity:** A standard network metric used to quantify the extent to which groups of nodes are more connected within themselves than to other groups [58]. It is based on node

partitions (or sets of communities) and given by

$$Mod = \sum_c \left( \frac{L_c}{m} - \epsilon \left( \frac{k_c}{2m} \right)^2 \right), \tag{2}$$

where $c$ is the index of the community or module, $m$ is the total number of edges, $L_c$ is the number of intra-community links, and $k_c$ is the sum of degrees of nodes in the community. The parameter $\epsilon$ is the resolution parameter, chosen to be 1. We manipulated networks using the python package NetworkX [59]. Optimal community partitions for modularity calculation were obtained using the Louvain greedy optimisation algorithm [60].

**Effective increase.** The network metrics described above are presented as their effective increase. We calculated them by subtracting the value obtained on the studied network for a given property from the value obtained for the corresponding Erdős–Rényi random network with the same size $S$ and connectance $C$. Thus, the effective increase of a metric $Z$ is $Z_s = Z - Z_r$, where $Z_r$ is the average value of the metric for the random network. In this way, we were interested in the increase from what would be expected by chance. For this calculation, we averaged over 50 random networks for each sample.

**Linear stability analysis.** To perform linear stability analysis on the resulting communities, we numerically calculated the Jacobian matrix of the system from Eq 1 at the equilibrium reached by the community after the occurrence of the last assembly event. The Jacobian is the matrix of derivatives of the right-hand side of Eq 1, defined as $J_{ij} = \frac{df_i}{dx_j}$, where $\frac{dx_i}{dt} = f_i$. We obtained the eigenvalues of this Jacobian matrix and assessed the stability of the system by representing the real and imaginary parts of the eigenvalues on a 2D plane.

## Supporting information

**S1 Fig. Evolved mutualism is a mechanism for the generation of consumer-resource interactions.** During each assembly event, a new successful (i.e. capable of growing and establishing) species is added to the community. *Evo.* scenarios are assembled via speciation, and *Inv.* scenarios are assembled via invasion. *No-M* scenarios do not feature mutualisms, while in *Inv. High-M* invading species have a proportion of mutualisms uniformly chosen between 0.8 and 1. The plots show the average values of 15 samples for every 50 assembly events (dots) and the vertical lines show the standard errors, calculated as the standard deviation divided by the square root of the number of samples. Each sample is a simulation of the entire assembly process for a scenario. The average number of consumer-resource interactions sustained by each species is higher when high proportions of mutualism are selected than when mutualism is not present. Evolution (i.e. speciation) enhances this effect, in comparison with invasion. Parameter values: interaction strengths were drawn from a half-normal distribution of zero mean and a standard deviation 0.2, and strength for consumers was made no larger than the strength for resources. Communities started with 5 non-interacting species. New species were given initial abundances equal to the extinction threshold $x_{ext} = 10^{-6}$. Connectance of invading species was drawn from a uniform distribution between 0.05 and 0.5. In speciation, offspring species had up to 5 interactions differing from the parent species, chosen randomly ($\Delta = 5$, see Methods). Handling times for mutualism and consumer-resource was 0.1. (PNG)

**S2 Fig. Mutualism drives the increase of degree entropy.** *Evo.* scenarios are assembled via speciation, and *Inv.* scenarios are assembled via invasion. *No-M* scenarios do not feature mutualisms, while in *Inv. High-M* invading species have a proportion of mutualisms uniformly chosen between 0.8 and 1. Degree entropy of communities normalised by the average

random network with the same richness and connectance. The random average was calculated using 50 random network samples. To calculate the entropy increase, the random average was subtracted from the degree entropy of all 15 community samples. Each sample is a simulation of the entire assembly process for a scenario. A value of zero increase corresponds to the average of the random counterparts, while positive values correspond to an increase from the value expected by chance. The degree entropy increases for all models, but a higher increase is driven by high proportions of mutualistic interactions. Mutualisms promote a higher homogeneity of degrees.
(PNG)

**S3 Fig. Evolution drives the increase of modularity.** *Evo.* scenarios are assembled via speciation, and *Inv.* scenarios are assembled via invasion. *No-M* scenarios do not feature mutualisms, while in *Inv. High-M* invading species have a proportion of mutualisms uniformly chosen between 0.8 and 1. Modularity of communities normalised by the average random network with the same richness and connectance. The random average was calculated using 50 random network samples. To calculate the entropy increase, the random average was subtracted from the degree entropy of all 15 community samples. Each sample is a simulation of the entire assembly process for a scenario. A value of zero increase corresponds to the average of the random counterparts, while positive values correspond to an increase from the value expected by chance. Modularity is mainly driven by speciation, with the lack of mutualism also being responsible for an increase. Invasion models with mutualism barely increase modularity, while evolution promotes the highest increase.
(PNG)

**S4 Fig. Results are robust across interaction strengths.** Reproduction of main results from Figs 1, 2, S2 Fig and S3 Fig for *Evo.* and *Inv.* scenarios with $\sigma$=(0.4,0.6,0.8), 5 samples each. As $\sigma$ gets larger, interactions become stronger. (A-H) All results remain similar for stronger interactions. Two notable differences are that 1. variability in connectance for evolution increases for stronger interactions and 2. entropy increase becomes smaller and modularity increase becomes larger for invasion. All parameter values are the same as in the main results.
(PNG)

**S5 Fig. *Evo.* results are consistent across inheritance degrees.** Reproduction of main results from Figs 1, 2, S2 Fig and S3 Fig for the Evo. scenario with $\Delta$=(15,25,35), 10 samples each. As $\Delta$ gets larger and the degree of inheritance gets lower, speciation results approach the ones obtained with assembly by invasion. However, a small inheritance still generates visible differences. (A-C) Richness gets lower, resulting in less complexity, but connectance gets higher at the beginning then remains at the same level in the end. (D-H) Interaction composition and network metrics change as predicted. All parameter values are the same as in the main results.
(PNG)

**S6 Fig. Assembly at different time-scales.** *Evo.* Reproduction of main results from Figs 1, 2, S2 Fig and S3 Fig for *Evo.* and *Inv.* scenarios with assembly events occurring before ecological equilibrium is attained, 5 samples each. The fastest assembly scenario is for 1 event at every 25 time-steps (1/25), then 1 at every 100 time-steps (1/100), and the slowest with 1 at every 1000 (1/1000), all with evolution or invasion happening in the same time-scale as the ecological dynamics. (A-H) All results remain the same as the ones obtained with separation of time-scales, apart from very fast evolution (Evo. 1/25). Composition of interaction types remains

similar, but complexity decreases considerably as a result of a much lower connectance. All parameter values are the same as in the main results.
(PNG)

**S7 Fig. *Evo.* patterns are marginally stronger when speciation is weighted by abundance.** *Evo.* Reproduction of main results from Figs 1, 2, S2 Fig and S3 Fig for an *Evo.* scenario in which the probability of a species being selected as a parent species for speciation is weighted by its abundance, instead of being the same for all species, 10 samples. (A-H) All results remain similar, with small variations. Richness grows to larger values, resulting in higher complexity. The selection for mutualistic interactions is also stronger. All parameter values are the same as in the main results.
(PNG)

## Author contributions

**Conceptualization:** Gui Araujo, Miguel Lurgi.

**Data curation:** Gui Araujo.

**Formal analysis:** Gui Araujo.

**Funding acquisition:** Miguel Lurgi.

**Investigation:** Gui Araujo, Miguel Lurgi.

**Methodology:** Gui Araujo, Miguel Lurgi.

**Project administration:** Miguel Lurgi.

**Resources:** Gui Araujo, Miguel Lurgi.

**Software:** Gui Araujo.

**Supervision:** Miguel Lurgi.

**Validation:** Gui Araujo, Miguel Lurgi.

**Visualization:** Gui Araujo.

**Writing – original draft:** Gui Araujo, Miguel Lurgi.

**Writing – review & editing:** Gui Araujo, Miguel Lurgi.

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
