## [Decision Letter · Decision Letter 0]

7 Mar 2025

PCOMPBIOL-D-24-02213

Mutualism provides a basis for biodiversity in eco-evolutionary community assembly

PLOS Computational Biology

Dear Dr. Lurgi,

Thank you for submitting your manuscript to PLOS Computational Biology. After careful consideration, we feel that it has merit but does not fully meet PLOS Computational Biology's publication criteria as it currently stands. Therefore, we invite you to submit a revised version of the manuscript that addresses the points raised during the review process.

Please submit your revised manuscript within 60 days May 07 2025 11:59PM. If you will need more time than this to complete your revisions, please reply to this message or contact the journal office at ploscompbiol@plos.org. Please include the following items when submitting your revised manuscript:

We look forward to receiving your revised manuscript.

Kind regards,

Youhua Chen

Academic Editor

PLOS Computational Biology

Zhaolei Zhang

Section Editor

PLOS Computational Biology

**Additional Editor Comments:**

Both reviewers felt the paper was interesting and important, and provided specific comments for the authors to revise. I have additional remarks for the authors: since the diverse interactions were important for network complexity and stability in your odel, provided the details on the model parameter configurations, for example, those parameters like interaction parameters c_ij in the general LV model. How and why were some specific parameter values used in your simulations? Moreover, can you conduct extensive parameter space searching to check the robustness and consistency of your results and conclusions? I think these were important to reproduce, at least making the results become more convincing.

**Journal Requirements:**

At this stage, the following Authors/Authors require contributions: Gui Araujo, and Miguel Lurgi. Please ensure that the full contributions of each author are acknowledged in the "Add/Edit/Remove Authors" section of our submission form.

3) We note that your Data Availability Statement is currently as follows: "All relevant data are within the manuscript and its Supporting Information files. If accepted for publication the computer code implementing the model and simulations will be made freely available on an open access code repository such as GitHub.". Please confirm at this time whether or not your submission contains all raw data required to replicate the results of your study. Authors must share the “minimal data set” for their submission. PLOS defines the minimal data set to consist of the data required to replicate all study findings reported in the article, as well as related metadata and methods (https://journals.plos.org/plosone/s/data-availability#loc-minimal-data-set-definition).

- The points extracted from images for analysis..

4) Please ensure that the funders and grant numbers match between the Financial Disclosure field and the Funding Information tab in your submission form. Note that the funders must be provided in the same order in both places as well.

**Reviewers' comments:**

Reviewer's Responses to Questions

**Comments to the Authors:**

Reviewer #1: The author considered generalized LV models of community assembly dynamics with mutualism, competition, predation, and evolution (speciation) and conducted numerical simulations. They found that highly mutualistic communities have complex and stable networks, causing a positive complexity-stability relationship. Because of the potential importance of eco-evolutionary dynamics, this kind of study is of interest for biologists. However, I feel the manuscript has some weaknesses.

Major comments:

1. Here "evolution" is equivalent to "speciation", but other types of evolutionary dynamics can affect community assembly. For example, Kondoh (2003) Science showed adaptive foraging can produce a positive complexity-stability relationship. It would be great if the authors can add discussion on the potential difference between speciation and other evolutionary dynamics (adaptive trait changes that result in interaction changes, genetic drift, etc.).

2. In the simulations, the authors assumed Delta = 5, but I was wondering how it affects the results. By changing the value of Delta, can the authors show the gradual change from "Evo." to "Inv."?

3. The difference between "Evo." and "Inv." is quite interesting. It would be great if the authors can add more discussion on the realistic scenario of "Evo." dynamics. Oceanic islands may produce community dynamics like "Evo." due to the scarcity of invasion events, but the timescale is a problem. Recent studies on eco-evolutionary dynamics consider rapid contemporary evolution, but speciation is usually excluded (e.g., Yamamichi et al. 2023 Ecol. Lett.). Thus, it will be interesting to check the timescale of speciation events in oceanic islands and the prevalence of mutualistic interactions in such isolated communities.

Minor comments:

Figure 1 legend: Does "richness" in (A) represent the number of species? It would be better to write the definition of "connectance" in the legend as well.

Figure 2A: It would be better to move the legend down a bit to avoid the overlap with the red lines.

Figure 2 legend: It will be nice to clarify that the green and yellow lines are overlapping in Fig. 2A.

Figure 3: I was wondering whether the authors really need colors here or not.

L431: have a differ from the parent a number → have a difference from the parent in a number

Reviewer #2: The authors introduce an eco-evolutionary assembly model, where ecological communities emerge either via speciation or immigration events (or a mix of both). The key novelty of the study is the mix of interaction types, meaning that assemblages consist of antagonistic, competitive and mutualistic interactions, whereas previous studies typically focused on single interaction types in isolation (such as food webs or pollination networks). The authors demonstrate that evolution promotes higher proportions of mutualism and that mutualistic interactions in turn enhance the dynamical stability of assembled communities. Overall, I truly enjoyed reading this manuscript. It is very well written, the story follows a clear logic and the results are highly interesting. I think it is a good fit for PLOS Computational Biology. My comments are mostly minor and listed below in chronological order.

- line 17-20: I don't understand this statement. Yes, pairwise mutualistic interaction create positive feedback, but the same is true for pairwise competitive interactions. So why should a higher proportion of competitive interactions favour stability?

- line 44-46: Maybe split the list of references in this sentence, so it becomes clear which ones refer to mutualistic systems and which ones refer to food webs. Also, consider adding the following references:

Becker, L., Blüthgen, N., & Drossel, B. (2022). Stochasticity leads to coexistence of generalists and specialists in assembling mutualistic communities. The American Naturalist, 200(3), 303-315.

Metz, T., Blüthgen, N., & Drossel, B. (2023). Shifts from non‐obligate generalists to obligate specialists in simulations of mutualistic network assembly. Oikos, 2023(7), e09697.

Allhoff, K. T., Ritterskamp, D., Rall, B. C., Drossel, B., & Guill, C. (2015). Evolutionary food web model based on body masses gives realistic networks with permanent species turnover. Scientific reports, 5(1), 10955.

- lines 48-55, as well as lines 75-77: This reminded me of the study by Morris et al. (2021), which also focusses on network assembly via a mix of mutations/speciations and invasions. Maybe it's worth referencing?

Morris, J. R., Allhoff, K. T., & Valdovinos, F. S. (2021). Strange invaders increase disturbance and promote generalists in an evolving food web. Scientific Reports, 11(1), 21274.

- line 85-87, as well as lines 181-184: I was very surprised by the finding that mutualism acts as a stabilising force. As mentioned above, mutualism creates positive (=self-reinforcing) feedback, so how can it be stabilising? As far as I know, systems actually tend to become LESS stable when mutualism outweighs antagonism (see for example Yacine and Loeuille 2022). Is this due to the functional response? Please clarify.

Yacine, Y., & Loeuille, N. (2022). Stable coexistence in plant-pollinator-herbivore communities requires balanced mutualistic vs antagonistic interactions. Ecological Modelling, 465, 109857.

- Along similar lines, Maliet et al (2020) showed that antagonistic interactions tend to foster species and trait diversity, while mutualistic interactions tend to reduce diversity, which seems to be in contrast to the key results presented here. Why these contrasting results? I think this apparent contradiction deserves to be discussed.

Maliet, O., Loeuille, N., & Morlon, H. (2020). An individual‐based model for the eco‐evolutionary emergence of bipartite interaction networks. Ecology Letters, 23(11), 1623-1634.

Please clarify.

- Fig 1: Why is there no "evo high M" scenario? Could that be implemented as well?

- Fig 1 and 2: Consider refering to line colous in the text. Also, please check whether your choice of colours works for colour-blind people.

- The error bars in most of the figures are very small, indicating that a given parameter combination would always lead to more or less the same network structure. Is this correct? If yes, is it possible to visualise these networks? How do they look like?

- caption of Fig 6: I think there is a typo: Mixed 0.2 should be 20% invasion, right?

- The discussion provides an excellent overview of the key results and their implications. However, I sometimes miss some information of how these results emerge from the underlying model assumptions. In general, I am not a big fan of putting the methods at the end (but I guess this is a Journal requirement?). I found it difficult to understand the results while reading the manuscript for a first time, and had to read it a second time, now going back and forth between methods and results section, before it finally made sense. A bit more info about the methods earlier in the manuscript might help.

- line 305: another typo: "Swueis" should be "Suweis"

- line 483: I'm not sure I understand why you made pij+=pij- if pij+>pij-. Doesn't this translate into the absence of convergence efficiency losses, so that every bit of resource biomass that is consumed turns into predator biomass?

- lines 400-408: I am not sure I understand this correctly. So a given species that is perfectly viable with a given set of interactions can be pushed to towards extinction by adding more positive interactions? Why? Why should the species suddenly suffer when it gets even more resources? From a technical perspective, I see that this assumption is probably needed to avoid ever growing networks, but from an ecological perspective I think it is problematic and needs additional clarification.

- line 414: you assume that the system reaches a steady state before the next species is added, which basically reflects a seperation of ecological versus evolutionary timescales. What is the reasoning behind this assumption? Is it necessary? How would the results change when overlapping timesclapes are taken into account? I'm asking because there is ample evidence for rapid evolution nowadays, see for example:

Hairston Jr, N. G., Ellner, S. P., Geber, M. A., Yoshida, T., & Fox, J. A. (2005). Rapid evolution and the convergence of ecological and evolutionary time. Ecology letters, 8(10), 1114-1127.

Thompson, J. N. (1998). Rapid evolution as an ecological process. Trends in ecology & evolution, 13(8), 329-332.

- line 422: Shouldn't the mutation rate somehow depend on biomass densities? I guess that larger populations have a higher probability to generate mutants, right?

- line 430-432: Check grammar in this sentence.

- line 497: As far as I know network modularity is usually calculated via simulated annealing. Is this also done in the python package that you use here? If yes, maybe explain a bit how that works. If not – why not using simulated annealing?

- line 506: typo "Erdos-Renyi" should be "Erdős–Rényi". Also, a reference is missing here.

**Have the authors made all data and (if applicable) computational code underlying the findings in their manuscript fully available?**

Reviewer #1: Yes

Reviewer #2: **No: **The authors state that the code will be made available if the manuscript is accepted, but so far it is not available.

PLOS authors have the option to publish the peer review history of their article (what does this mean?). If published, this will include your full peer review and any attached files.

Reviewer #1: No

Reviewer #2: No

**Figure resubmission:**
---

## [Decision Letter · Decision Letter 1]

17 Jul 2025

PCOMPBIOL-D-24-02213R1

Mutualism provides a basis for biodiversity in eco-evolutionary community assembly

PLOS Computational Biology

Dear Dr. Lurgi,

Thank you for submitting your manuscript to PLOS Computational Biology. After careful consideration, we feel that it has merit but does not fully meet PLOS Computational Biology's publication criteria as it currently stands. Therefore, we invite you to submit a revised version of the manuscript that addresses the points raised during the review process.

Please submit your revised manuscript within 30 days Sep 16 2025 11:59PM. If you will need more time than this to complete your revisions, please reply to this message or contact the journal office at ploscompbiol@plos.org. Please include the following items when submitting your revised manuscript:

We look forward to receiving your revised manuscript.

Kind regards,

Youhua Chen

Academic Editor

PLOS Computational Biology

Zhaolei Zhang

Section Editor

PLOS Computational Biology

**Additional Editor Comments:**

Dear Author: Thank you for your careful revision and both reviewers now had made additional but minor comments for you to improve. Please revise the paper throughout again before final acceptance.

**Reviewers' comments:**

Reviewer's Responses to Questions

**Comments to the Authors:**

Reviewer #1: The manuscript was improved through revision, but I still have some comments.

Abstract: It will be great if the authors can clarify that "an evolutionary process" is speciation and "a purely ecological assembly" is via migration here. Then readers will be able to understand what "in isolation" means easily.

Author summary: It may be better to avoid using "predator-prey" and "consumer-resource" as some readers may wonder the difference between them. What about removing "consumer-resource" and using "predator-prey" throughout the author summary?

L12: The term "eco-evolutionary" is sometimes very confusing due to the time-scales of dynamics as pointed out by Bassar et al. (2021) Towards a more precise – and accurate – view of eco-evolution. Ecology Letters. It will be helpful to clarify the term here.

L18-19, 32: Readers may wonder linear functional responses destabilize dynamics with mutualism whereas saturating functional responses stabilize dynamics. It would be better to clarify the relationship between functional responses and stability.

L32, 88: It might be helpful to clarify that "saturating" and "Type II functional responses" are equivalent.

L33: It will be nice to explain the term "priority effects" for readers who are not familiar with it.

L147: richness (A) -> richness (S) (A)

L148: connectance (B) -> connectance (C) (B)

L150: S and C should be in Italic.

L235: What does "n reps" mean?

L443-444: Did Kondoh (2003) consider genetic drift? How does drift promote the emergence of complexity in communities?

Reviewer #2: Once again, I very much enjoyed reading this very interesting manuscript. The previous version was already of very high quality but the manuscript nevertheless improved during peer review.

I have only one small remark on competitive feedback loops. The authors write in their response that "competitive interactions do not create a positive feedback". I disagree. Competitive interactions (in isolation) always result in positive (=self-reinforcing) feedback, simply because the product of two negative numbers is a positive number. For clarification, imagine two competing populations at equilibrium. If species 1 experiences a decline in abundance due to an external disturbance, then this decline will lead to a reduction in competitive pressure on species 2, so species 2 can reach higher abundances, which will in turn translate into an increased competitive pressure on species 1, thus reinforcing the initial disturbance. In the absence of all other forces, this feedback loop is thus clearly destabilising, because it pushes species 1 towards extinction and species 2 towards high abundances.

However, other forces, such as selfregulation via intraspecific competition and/or interactions with other species, will most likely prevent too high (and hence unrealistic) abundances of species 2. In general, I would argue that it is not very insightful to look at single feedback loops in isolation, given that feedback loops usually act in concert and potentially balance each other. For more context, I recommend to read the following paper: Levins, R. (1974). Discussion paper: the qualitative analysis of partially specified systems. Annals of the New York Academy of Sciences, 231(1), 123-138.

Plase note that I bring this up in response to the author's response, but I don't think this requires additional revision of the text. In fact, all my comments have been thoroughly adressed. I do not have any further suggestions for improvement and recommend acceptence of the article as is. I thank the authors for their extensive and insightful replies!

**Have the authors made all data and (if applicable) computational code underlying the findings in their manuscript fully available?**

Reviewer #1: Yes

Reviewer #2: Yes

PLOS authors have the option to publish the peer review history of their article (what does this mean?). If published, this will include your full peer review and any attached files.

Reviewer #1: No

Reviewer #2: No

**Figure resubmission:**
---

## [Editor Report · Decision Letter 2]

7 Aug 2025

Dear Dr Lurgi,

We are pleased to inform you that your manuscript 'Mutualism provides a basis for biodiversity in eco-evolutionary community assembly' has been provisionally accepted for publication in PLOS Computational Biology.

Best regards,

Youhua Chen

Academic Editor

PLOS Computational Biology

Zhaolei Zhang

Section Editor

PLOS Computational Biology

Dear author, I am pleased to recommend the publication of your interesting paper, congratulations!

---

## [Editor Report · Acceptance letter]

PCOMPBIOL-D-24-02213R2

Mutualism provides a basis for biodiversity in eco-evolutionary community assembly

Dear Dr Lurgi,

I am pleased to inform you that your manuscript has been formally accepted for publication in PLOS Computational Biology. Your manuscript is now with our production department and you will be notified of the publication date in due course.

With kind regards,

Benedek Toth
